# The Dark Side of Process Mining in Public Organizations: Data Access Constraints as Evidence of Organizational Opacity

**Abstract.** Obtaining an analytically usable event log is a prerequisite for process mining; however, what the literature rarely documents are the organizational, technical, and epistemic conditions that determine what becomes accessible and analytically usable in the first place. This paper argues that event logs are organizational artifacts: digital footprints that encode, in their completeness or incompleteness and their institutional accessibility or unavailability, the conditions under which an organization records and relates to its own processes. Drawing on first-hand research experience applying process mining to litigation data from a large public legal organization, a context systematically underrepresented in the process mining literature, we encountered constraints spanning institutional access, technical infrastructure, and the semantic architecture of the recording system across a dataset of 14,661,857 events. Drawing on organizational opacity theory, we propose an exploratory typology of three dimensions of processual visibility, institutional, technical-structural, and epistemic, and show that each corresponds to a distinct failure in the visibility architecture that process mining presupposes. The typology functions as a conceptual instrument for identifying and comparing data access constraints across organizational contexts, and positions their systematic documentation as a legitimate scientific contribution rather than methodological failure.

Keywords: Process mining · Public sector organizations · Data access constraints · Organizational opacity · Processual visibility · Event logs as organizational artifacts.

## 1 Introduction

Process mining projects spend a disproportionate share of their effort before the analysis even begins. Extracting event data from heterogeneous sources [1, 2], correlating events without case identifiers [3, 4], and preparing logs for analysis [5] are recognized challenges, and the literature has responded with a growing body of technical solutions. What it has not responded to is a simpler and more uncomfortable question: what actually happens in practice, inside real organizations, when researchers try to obtain the data they need? That experience, the negotiations, the format incompatibilities, the semantic gaps, the activities that leave no trace, rarely makes it into the published record. It is absorbed into a footnote, or dropped entirely. This paper argues that it should not be.

We report on a first-hand research experience applying process mining to litigation data from a large public legal organization, in a dataset of 14,661,857 events across 451,822 cases recorded between March 2022 and February 2026. What we encountered

was not a single obstacle but a layered set of conditions: a file format incompatible with standard process mining tools, requiring direct parsing in R; 48,969 distinct activity labels that were unstandardized, abbreviation-laden, and semantically inconsistent with the organization's own normative process model; and after an intensive harmonization effort combining fuzzy string matching and keyword-based categorization, 19.7% of events that remained unmapped, not because the method failed, but because those activities were executed outside the formal system and left no digital trace. These barriers were not incidental; they were patterned, and their pattern said something about the organization that no process map could capture.

The process mining literature has not engaged with this kind of experience as a research object. Existing work addresses adjacent challenges, including privacy and confidentiality constraints on event log release [6], technical imperfections in event log quality [7], and organizational factors shaping process mining adoption [8], but these contributions assume the data exists and ask what to do with it. Empirical studies that have successfully applied process mining in the public sector tend to come from contexts with mature information systems and institutionalized data governance practices [10]. The organizational conditions that make data access difficult, costly, or structurally incomplete, and what those conditions reveal about the institutions that produce them, remain unexamined. Martin et al. [9] identified data access constraints as a persistent challenge in process mining practice; this paper takes that observation seriously and asks what it looks like from the inside.

The contributions of this paper are three. First, drawing on a concrete research experience in a large public legal organization, it reframes data access constraints as empirical findings in their own right rather than pre-analytical noise to be overcome before the real work begins, and argues that their systematic documentation is a legitimate and enabling scientific contribution. Second, it proposes an exploratory typology of three dimensions of processual visibility, institutional, technical-structural, and epistemic, grounded in that experience and going beyond the technical framing dominant in the data quality literature to incorporate the organizational and epistemic conditions that shape what data becomes available for analysis in the first place. Third, it connects process mining research to organizational opacity theory, interpreting the characteristics of an event log as symptoms of how an organization relates to its own processes, and opening a dialogue between two bodies of literature that have not previously been in conversation.

The remainder of this paper is organized as follows. Section 2 reviews relevant background. Section 3 presents the case. Section 4 proposes the typology. Section 5 develops the organizational opacity lens. Section 6 discusses implications and opens a research agenda. Section 7 concludes.

## 2        Background

### 2.1        Process Mining in Public Sector Organizations

Process mining enables the objective reconstruction of how organizational processes actually unfold by extracting knowledge from event logs recorded in information systems [5]. In public sector organizations, this capability is particularly consequential: the gap between prescribed and enacted behavior has direct implications for legal compliance, accountability to citizens, and the quality of public service delivery [10].

Empirical applications in the public sector have been documented predominantly in European contexts where enterprise information systems are mature and data governance practices are institutionalized. These conditions, standardized process models, accessible event logs, and organizational readiness, are rarely made explicit as prerequisites, yet they are systematically present in published success cases. Klievink et al. [11] found that public sector bodies may be technically capable of processing large datasets but frequently lack the organizational fit and institutional structures to derive value from them. This finding anticipates, without naming, the data access problem that process mining researchers encounter in less mature contexts.

The organizational and managerial dimensions of adoption remain underexplored: Grisold et al. [8] and vom Brocke et al. [12] both note that neither algorithmic advances nor adoption frameworks address whether analytically usable event data exists in the first place. That prior challenge is only partly addressed by the data quality literature. Suriadi et al. [13] propose a systematic taxonomy of event log imperfection patterns and frame data quality as a technical problem to be diagnosed and corrected before analysis begins. Ter Hofstede et al. [7] extend this framing, arguing that process-data quality is the true frontier of the field. More recently, Bartelheimer et al. [14] demonstrate that workarounds generate systematic mismatches between event logs and real-world process execution, and del Rio Ortega et al. [15] consolidate this trajectory in a BISE special issue. What these contributions share is a technical framing: data quality is treated as a property of logs that already exist. None addresses the prior condition in which logs are structurally inaccessible, semantically impenetrable, or institutionally withheld, nor what that condition reveals about the organization that produced them.

### 2.2        Organizational Opacity and the Limits of Visibility

The assumption that digital systems produce transparency is pervasive in both practice and scholarship. Stohl et al. [16] challenge this assumption by distinguishing between visibility, the combination of availability, approval, and accessibility of information, and transparency. They argue that visibility and transparency are distinct concepts and neither is straightforwardly associated with accountability. High levels of visibility can, paradoxically, produce what they term organizational opacity: a condition in which information is nominally present but structured, formatted, or distributed in ways that render it impenetrable to meaningful analysis.

Stohl et al. [16] identify two primary mechanisms through which organizational opacity is produced: strategic opacity, in which actors deliberately manage information

visibility to limit accountability, and inadvertent opacity, in which the architecture of information systems produces impenetrability as an unintended consequence of design choices, institutional inertia, or lack of coordination across systems. The barriers encountered in the case reported here are better understood as inadvertent, the accumulated result of a proprietary system designed for legal case management, not for process analysis; of activity labeling practices developed by clerks, not by data architects; and of a registration culture shaped by workflow imperatives rather than analytical accountability.

Taken together, these two bodies of literature reveal a shared blind spot: neither has examined what it means when an organization structurally cannot render its own processes legible. The following sections use the case reported here to fill that gap.

## 3     The Case: When Data Resists Analysis

This paper adopts an empirical-reflective approach grounded in a single case, drawing on a doctoral research project involving the extraction and analysis of event log data from the litigation management system of a large public legal organization. The barriers reported below were identified through retrospective analysis of field notes, technical logs, and correspondence records generated during the data access process. The study is positioned within the interpretive tradition of IS research, using the case as an instrument for conceptual development rather than as a basis for statistical generalization [17].

The organization is the legal representation body of a large and economically significant jurisdiction, with 89 organizational units and a caseload spanning hundreds of thousands of active disputes across fiscal, administrative, and judicial domains. Since 2022, its litigation workflows have been managed through a proprietary case management system developed specifically for the institution. The specific object of analysis is the general litigation process, responsible for all judicial proceedings involving the jurisdiction and its autarchies outside the tax-fiscal domain. Table 1 summarizes the barriers encountered across five dimensions.

Table 1. Data access constraints encountered in the case.

| Layer | Barrier | Concrete manifestation |
|---|---|---|
| Institutional access | Prolonged negotiation for data release | Dependence on an internal research unit as intermediary; no formal data access protocol |
| Technical extraction | Proprietary format of the case management system | Incompatibility with standard PM tools (ProM, Disco); direct parsing in R required |
| Volume and structure | ~14.6M events across 451K cases (4 GB) | Requires non-standard infrastructure and tooling |
| Semantic quality | Inconsistent labels, abbreviations, no controlled vocabulary | 80.3% coverage after Jaro-Winkler harmonization; 19.7% unmapped |
| Registration gaps | Activities executed outside the formal system | Workarounds that leave no trace in the log by definition |

*Source: Authors' own elaboration.*

These barriers are not independent. They reinforce one another and collectively reflect a condition that is common but rarely documented [9]: organizational systems are designed to serve operational purposes, not analytical ones. What the process of obtaining a usable event log cost in practice is difficult to quantify precisely, but its dimensions are clear. It required months of institutional negotiation, non-standard technical infrastructure, and a semantic harmonization effort that, even after completion, left nearly one in five events uninterpretable, not because the harmonization failed, but because those events correspond to activities the recording system was never designed to capture. The barriers did not prevent analysis entirely, but they shaped, constrained, and partially obscured what the analysis could see. That shaping effect is itself an empirical finding, and it is the subject of the typology proposed in the following section.

## 4      A Typology of Data Access Constraints in Process Mining

The five constraint layers documented in the case reveal a structured pattern that can be organized into three analytically distinct categories. We propose these categories as dimensions of processual visibility, each capturing a distinct way in which an organization's relationship with its own processes is encoded in the characteristics of the event log it produces. This typology goes beyond the technical framing dominant in the data quality literature [7, 13] to incorporate the institutional and epistemic dimensions that shape what data becomes available for analysis in the first place.

**Type I: Institutional Constraints.** Institutional barriers and the absence of formal data access protocols are documented as a persistent challenge [18, 9]. In the case reported here, this barrier was partially mitigated by the existence of an institutionalized internal research unit. Without such intermediary structures, access would likely have been unfeasible within an academic research timeline.

These barriers reflect how the institution conceives of its data as an internal operational resource rather than a public or research asset. Goel et al. [18] demonstrate the need for dedicated data governance frameworks for process data, arguing that governance deficits, including the absence of formal access protocols and organizational accountability for process data, directly undermine the reliability of process mining insights. Martin et al. [9] found that governance, people, and culture are the domains where organizations identify the most challenges in process mining adoption. Their consequence for research is severe: institutional constraints make replicability structurally difficult, and they introduce a selection effect into the literature, only researchers with the right institutional connections can conduct this type of study.

**Type II: Technical-Structural Constraints.** Technical-structural barriers concern the infrastructure of data recording and storage. Resolvability is assessed in terms of engineering effort and resource availability: unlike institutional constraints, technical-structural constraints can in principle be overcome through investment in infrastructure and custom extraction tooling.

In the case reported here, this category encompasses the proprietary format of the case management system, incompatible with process analysis export; the volume of the dataset, requiring non-standard infrastructure; and the absence of any native export

functionality oriented toward analytical use. These are not exceptional conditions: they are the norm in organizations whose systems were built for operational rather than analytical purposes, as Klievink et al. [11] document.

The critical point is not that these barriers exist, but that their resolution cost is insufficiently conveyed in the literature: papers that complete the analysis describe the transformation layer in terms that understate the time, expertise, and iterative effort involved, producing a literature in which technically demanding studies appear methodologically routine.

**Type III: Epistemic Constraints.** Epistemic barriers are the most consequential and the least visible. They concern not the access to data or its technical form, but what the data can and cannot represent: the gap between what an organization records and what actually happens in its processes. Resolvability is very low, epistemic constraints are constitutive of the organizational context in which the recording system was designed.

In the case reported here, epistemic constraints manifested in two forms. The first is semantic opacity: 48,969 distinct activity labels with no controlled vocabulary, reflecting a recording culture in which activities originate from two structurally different sources, an external judicial system with standardized automated labeling, and internal staff with manual and highly heterogeneous entry practices. An intensive harmonization effort combining fuzzy string matching and keyword-based categorization achieved 80.3% coverage. A formal meeting with a domain specialist confirmed that the remaining gap reflects a structural condition: a portion of those events corresponds to activities genuinely absent from the normative process model, inserted manually by practitioners to record informal steps the system was not designed to capture.

The second form is structural invisibility: activities executed outside the formal system leave no trace in the log by definition. Outmazgin and Soffer [19] demonstrate precisely this limitation, workarounds enacted through informal channels are, from the event log's perspective, non-existent, and process mining can only detect deviations that leave a digital trace.

Epistemic barriers define the analytical horizon of what process mining can see. A log that is 80.3% semantically mapped is not a partially clean dataset awaiting further processing; it is a representation of a process in which 19.7% of enacted behavior is structurally illegible. Table 2 summarizes the three constraint types.

Table 2. Typology of data access constraints in process mining.

| Type | Locus | Resolvability basis | Resolvability | Consequence for research |
|---|---|---|---|---|
| I – Institutional [18, 9] | Governance and power structures | Policy and cultural change required | Low | Limits replicability; selection bias |
| II – Technical-structural [11] | Infrastructure and systems | Engineering effort, resources | Moderate | Increases cost; understated in output |
| III – Epistemic [19] | Recording culture and system design | Requires redesigning what the system records | Very low | Defines analytical horizon |

*Source: Authors' elaboration, 2026.*

# 5      Organizational Opacity as an Analytical Lens

## 5.1      From Constraints to Symptoms

The typology proposed in Section 4 describes what researchers encounter when they attempt to apply process mining in a public sector organization with low data maturity. But description alone is insufficient. This section argues that the three constraint types, taken together, are not a collection of independent obstacles; they are symptoms of a single underlying condition: organizational opacity.

Stohl et al. [16] define opacity as a condition in which the architecture of information systems produces impenetrability rather than accountability. Their concept of inadvertent opacity can be extended to the case reported here, where impenetrability arises not from strategic concealment but from the accumulated effect of design choices, institutional inertia, and lack of coordination across systems that were never intended for analytical use.

## 5.2      Mapping the Typology onto the Opacity Framework

Stohl et al. [16] conceptualize visibility as the combination of three attributes: availability of information, approval to disseminate it, and accessibility to third parties. Table 3 shows how the three constraint types correspond directly to these attributes.

Type I Institutional barriers concern approval: data exists within the system but is not formally available to external parties. The absence of a data access protocol means that the approval dimension is governed by informal relationships rather than institutional rules.

Type II Technical-structural barriers concern accessibility: data may be nominally available and approved for release, but its format, volume, and infrastructure requirements render it inaccessible to all but the most technically resourced teams.

Type III Epistemic barriers concern availability itself: a substantial portion of what happens in the organization's processes is simply not recorded. Information that does not exist in the system cannot be made available, approved, or accessed. This is the deepest layer of opacity.

Table 3. Mapping constraint types onto the organizational opacity framework.

| Barrier type | Visibility attribute [16] | Opacity mechanism | Example in the case |
|---|---|---|---|
| I – Institutional | Approval | Inadvertent: no formal access protocol | Access mediated by relationships, not rules |
| II – Technical-structural | Accessibility | Inadvertent: proprietary format, volume | Dataset incompatible with standard tools |
| III – Epistemic | Availability | Constitutive: recording architecture | 19.7% unmapped; informal activities leave no trace |

*Source: Authors' elaboration based on Stohl et al. [16].*

### 5.3    Two Implications

This interpretation has implications at two levels. At the methodological level, it re-frames what process mining researchers encounter in contexts like the one reported here. The barriers are not pre-analytical noise; they are data. An organization's inability to produce an analytically usable event log is itself evidence of how that organization is structured, governed, and how it relates to its own process knowledge. Reporting these barriers systematically is therefore not a consolation prize for a study that could not fully execute its design; it is a legitimate empirical contribution and, critically, an enabling one.

At the organizational level, the opacity lens has diagnostic potential. Rosemann and vom Brocke [20] identify governance, information technology, people, and culture as core elements of BPM capability, precisely the dimensions along which the three constraint types are distributed. If the quality of an event log reflects the visibility architecture of the organization that produced it, then log quality can function as a proxy indicator of BPM maturity in a deeper sense than tool adoption alone.

## 6        Discussion and Research Agenda

The findings of this paper are in dialogue with three bodies of prior work. The first concerns the organizational and managerial challenges of process mining adoption. Martin et al. [9] produced a structured inventory of 32 challenges for the use of process mining in organizations. Two are directly instantiated by the case reported here: C.4 (data access constraints limiting PM across departmental and organizational boundaries) and C.9 (unavailability of event data needed for PM). Notably, Martin et al. [9] observe that C.4 had already been identified as a prominent limitation by Claes and Poels (2013) and remains unresolved over a decade later. The present paper documents at the level of research practice what that expert perception entails.

The second body concerns data quality in process mining. The case confirms the imperfection patterns described by Suriadi et al. [13], but fundamentally extends the framing of Ter Hofstede et al. [7]. The root of the quality problem is not technical: it is institutional and epistemic. The 48,969 distinct activity labels are not a data cleaning challenge; they are the artifact of a recording culture never designed for analytical use. This distinction, between quality as a technical property of existing data and quality as a consequence of how recording systems are designed, constitutes the primary conceptual contribution of the typology in Section 4. Bartelheimer et al. [14] move in this direction; the present paper extends this by interpreting such mismatches as organizational, not merely technical, phenomena.

The third body concerns process mining in the public sector. Racis and Spano [10], like most empirical work in government contexts, presuppose that data is available. The case shows that in low data maturity contexts the challenge is not adoption but the structural condition that makes access itself uncertain, costly, and non-replicable.

The selection effect deserves particular attention, because when structural barriers are severe enough to prevent analysis, the study tends not to be published, not for lack of value, but because the field offers no category for receiving an unproduced analysis

as a contribution. Over time, the record accumulates almost exclusively cases that worked. This bias is hard to detect because its instances are the studies that were never written, an absence invisible to any survey of what exists. The present paper is itself evidence: the barriers documented here were nearly absorbed into a footnote before being recognized as warranting separate treatment.

**Limitations.** The typology is exploratory and derived from a single case; generalization requires comparative investigation. The opacity interpretation is a theoretical proposition, not an established finding. The barriers were encountered sequentially, and their retrospective organization reflects analytical judgment rather than prospective design.

**The typology as a reporting framework.** Beyond its function as a conceptual instrument, the typology can serve as a first proposal for how process mining studies should report data access conditions. The field currently lacks shared norms for documenting what happens before the analysis begins. A study that characterizes its data access conditions along the three dimensions, institutional, technical-structural, and epistemic, provides information directly useful to other researchers and practitioners. This paper is itself an application of that framework.

**Research Agenda.** The analysis opens four questions. (1) How can data access protocols be developed for public sector organizations that enable process mining research without compromising privacy or legal confidentiality? (2) Can event log quality function as a measurable indicator of BPM maturity, following Rosemann and vom Brocke [20]? (3) How can the field develop mechanisms, venues, and reporting norms that counter the selection effect, so that structurally constrained studies enter the record? (4) What methodological adaptations are needed for process mining in contexts where event log incompleteness is structural rather than incidental?

# 7    Conclusion

This paper reported on and structurally analyzed the obstacles encountered in obtaining an analytically usable event log from a large public legal organization, documenting five constraint layers organized into an exploratory typology of three dimensions of processual visibility: institutional, technical-structural, and epistemic. Interpreted through the lens of organizational opacity theory [16], each constraint type corresponds to a distinct failure in the visibility architecture of approval, accessibility, and availability that shapes what data becomes available for analysis before any algorithm runs. The typology is offered both as a conceptual instrument for future comparative research and as a first proposal for how process mining studies can report data access conditions in a structured and communicable way.

Future work should pursue four directions: developing formal data access protocols for public sector process mining partnerships; testing whether event log quality can serve as a proxy for BPM governance maturity; investigating the selection effect that renders invisible the studies that encountered structural data access barriers; and developing methodological adaptations for process mining in contexts where event log incompleteness is structural rather than incidental.

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

## Appendix A: History of This Paper

This paper did not begin as a paper. It began as a problem.

The research reported here emerged from a doctoral project on organizational workarounds in public legal processes, which sought to apply an established workaround detection framework to event log data from a large public legal organization. The analytical ambition was straightforward: obtain the data, run the analysis, detect the workarounds. What followed was months of institutional negotiation, format incompatibilities, infrastructure limitations, and a harmonization effort that, even after completion, left nearly one in five events semantically unmapped.

At some point during that process, it became clear that the difficulties themselves were telling a story that the companion analysis, focused on workaround detection, would not be able to tell. The barriers were not incidental. They were patterned. And their pattern said something about the organization that no process map could capture.

The decision to write this paper separately was not a consolation for a study that had not gone as planned. It was a recognition that what had been encountered, and what had almost been silently absorbed into a methodological footnote, deserved to be named, documented, and submitted to a community equipped to receive it. A workshop with an explicit commitment to results that mainstream venues overlook seemed like exactly the right place.

The doctoral research continues. The workaround analysis proceeds with the data that was obtained. This paper reports on what that process cost and argues that the cost itself is evidence worth preserving.