# OpenReview forum: "The Dark Side of Process Mining in Public Organizations: Data Access Constraints as Evidence of Organizational Opacity"
_bpm-conference.org/BPM/2026/Workshop/FOR-BPM — FOR-BPM_

### Official Review · Reviewer_RmkA · 2026-06-22
**Constraints in process mining: data provenance and data lineage**

**Rating:** 4

**Review:**

This paper presents an interesting discussion on the difficulties of obtaining data for process mining analysis by discussing different types of challenges and corresponding sources of "organizational opacity". The paper proposes a framework of institutional, technical-structural, and epistemic constraints, and how these can be used to characterize how an organization records, governs, and understands its own processes. The paper is based on the authors' experience, which, while being a single anecdote, is certainly relatable to many other researchers in the community.

The paper is very relevant for FOR-BPM workshop and it has definitely the potential to spark relevant discussions and identify, for researchers, relevant research directions that can lead to actual value for organizations (rather than focusing on an "epsilon improvement" of the outcomes of an algorithm). In particular, the paper is valuable because it shifts attention from successful analysis outcomes to the actual conditions required for analysis.

What I found particularly noteworthy is the sentence "data quality is treated as a property of logs that already exist". This highlights that there are "unknown unknowns" that are discussed about. In doing so, the paper opens an important discussion about publication bias in BPM and process mining: we tend to see studies where event logs were obtained, cleaned, and analyzed, while failed, partial, or structurally challenged studies remain invisible. Such bias needs to be reconciled with the need for scientific evidence of the results, which is grounded in data and statistically sound outcomes.

The proposed typologies for data access constraints (institutional, technical-structural, and epistemic) are nicely grounded on the scientific literature and represent a possibly valuable framework for helping researchers describe what happened before analysis began, especially in public sector or low-data-maturity contexts where data access is complex, negotiated, and incomplete.

Possible improvement points are references to data provenance and data lineage. The paper shows that understanding an event log requires understanding where the data came from, who created it, under what institutional conditions, through which systems, and with what transformations. These are provenance and lineage questions. For process mining, these are not secondary technical details but central aspects to the interpretation of any discovered process model. Some results on visualization and interpretability could be brought up in the discussion.

The discussion about the constraint seems to put the 3 types at the same level. However, in my opinion, while institutional and technical constraints can be overcome, the epistemic constraints are much more fundamental. What I mean is that if I want to analyze data from last year, I can work towards fixing type I and II constraints but if type III is challenging my analysis, there is no way to overcome it. So, somehow, I see that the 3 constraint types have a "time direction" when it comes to resolvability.

I encourage the authors to add a short subsection (in the Discussion?) about data provenance and data lineage to make it more relatable to the data engineering field, beyond IS.

All in all, the paper represents a nice contribution that I would like to discuss during the workshop :)

**Advancing Bpm Thinking:**

I think the paper bst advance the BPM community thinking by explicitly suggesting to reflect not only on our process mining results, but by treating data access, incompleteness, and opacity not merely as practical obstacles, but as fundamental BPM phenomena: event logs are shaped by governance, technical infrastructures, recording practices, and organizational practices. This is an important shift because it prompts the BPM community to examine the conditions under which process knowledge has the possibility to emerge and become visible.

The paper implies that researchers need to know where event data comes from, how it was produced, who controlled access to it, how it was transformed, and what was lost or changed along the way. Making an explicit connection to data provenance and data lineage could help position the paper within neighboring research fields. In particular, the proposed typology could become a practical reporting framework for describing the provenance and lineage of event logs, BEFORE process mining analyses.

The paper could also benefit from a more in-depth discussion about the "so what?" aspects. As a technical person, I see the "resolvability basis" for type II, but those for type I and III remain a bit too vague for me.

In addition, I am induced to think that constraints I and III are not necessarily as isolated as the paper let me think. Could it be that a type III constraint is the result of a type I constraint? If so, the resolvability of the type III could be trivial, once the type I constraint is resolved. In other words, a more holistic treatment of the constraints could be useful to get a broader picture.

Finally, the paper targets "public organizations" but it would be interesting to understand to what extent the discussion on constraints is limited to public organizations. I do not think authors leverage their experience too much in any case (being just a single case), so I would like to read a more argumentative discussion on why authors believe that their framework applies only to public orgs.

---

### Official Review · Reviewer_5jCw · 2026-07-05
**Data quality through the opaque lens of data accessibility**

**Rating:** 4

**Review:**

This paper addresses the obstacles in obtaining a quality log, using a large law firm as an example. The paper proposes a 3D typology of constraints, namely institutional, technical-structural, and epistemic, each representing a different type failure in a clear recording of data.

**Advancing Bpm Thinking:**

The paper raises an imprortant question: how to tackle the problem of data accessibility in opaque situations. While the paper is not very clear in the way the problem is presented, i believe it will good to pose the question in the FOR-BPM workshop.

I recommend structuring the discussion along the data life-cycle, looking at ways to increase clarity, starting from design time (as proposed by the authors), continuing with processing time, and concluding with some iterative process with users to provide new suggestions to the next cycle of design.

---

### Official Review · Reviewer_4z1c · 2026-07-08
**The trouble of getting access data**

**Rating:** 4

**Review:**

The paper presents the experiences of the authors with getting access to data for process mining. They use a large case study where they went through the struggles of obtaining data from a public body to ground the various phases of such a project and the challenges faced along the way in existing research.

The paper is very insightful. It provides a nice overview of the true challenges of performing process mining projects in public sector organization. Now, I feel that it is important to realize that the paper does not clearly distinguishes between 'public-domain projects', such as research or student projects by an academic institute and 'private-sector projects', such as commercial projects by a process mining vendor.

This distinction is a bit problematic and food for discussion. For example, the authors refer to Stohl [16] to say that opacity is either strategic, where data is withheld purposely or inadvertent, where data is withheld because of architectural problems. However, while I believe this may be true for commercial projects, there is a third category for public-domain projects, namely the risk of reputation damage if the data or the results of the analysis are published and made publicly available. I would not categorize this as deliberately managing information to limit accountability, but also not as inadvertent opacity, i.e. a third category is needed here.

In Table 3, suddenly the term 'constitutive opacity' is introduced which does not match the 'strategic opacity' mentioned before. On top of that both Institutional and Technical-structure barriers may have opacity that are not inadvertent, but more related to the risk of reputation damage. For example, publishing data from specific systems may disclose that the organization uses these systems.

Overall, the work in the paper is executed in a methodologically sound way and the conclusions are grounded in literature (and I can relate to them from personal experience with process mining projects in governmental organizations). I therefore believe this paper is definitely worth discussing at a workshop.

**Advancing Bpm Thinking:**

I think the authors should extend Sections 5.3 and/or 6. More research is needed to provide methods, tools and techniques to overcome reputational challenges when starting process mining projects in the public domain. This should not be solves by confidentiality agreements (as you would in commercial projects).